# The Importance of Being Versatile: INFN-CHNet MA-XRF Scanner on Furniture at the CCR "La Venaria Reale"

Leandro Sottili [1,2], Laura Guidorzi [1,2], Anna Mazzinghi [3,4], Chiara Ruberto [3,4], Lisa Castelli [4], Caroline Czelusniak [4], Lorenzo Giuntini [3,4], Mirko Massi [4], Francesco Taccetti [4], Marco Nervo [5], Stefania De Blasi [5], Rodrigo Torres [6], Francesco Arneodo [6], Alessandro Re [1,2,*] and Alessandro Lo Giudice [1,2]

1   Dipartimento di Fisica, Università degli Studi di Torino, Via Pietro Giuria 1, 10125 Torino, Italy; leandro.sottili@unito.it (L.S.); laura.guidorzi@unito.it (L.G.); alessandro.logiudice@unito.it (A.L.G.)
2   Istituto Nazionale di Fisica Nucleare (INFN), Sezione di Torino, Via Pietro Giuria 1, 10125 Torino, Italy
3   Dipartimento di Fisica e Astronomia, Università degli Studi di Firenze, Via Giovanni Sansone 1, Sesto Fiorentino, 50019 Firenze, Italy; anna.mazzinghi@unifi.it (A.M.); ruberto@fi.infn.it (C.R.); lorenzo.giuntini@unifi.it (L.G.)
4   Istituto Nazionale di Fisica Nucleare (INFN), Sezione di Firenze, Via Giovanni Sansone 1, Sesto Fiorentino, 50019 Firenze, Italy; castelli@fi.infn.it (L.C.); czelusniak@fi.infn.it (C.C.); massi@fi.infn.it (M.M.); taccetti@fi.infn.it (F.T.)
5   Centro Conservazione e Restauro "La Venaria Reale", Piazza della Repubblica, Venaria Reale, 10078 Torino, Italy; marco.nervo@centrorestaurovenaria.it (M.N.); stefania.deblasi@centrorestaurovenaria.it (S.D.B.)
6   Division of Science, New York University Abu Dhabi, P.O. Box, Saadiyat Island, Abu Dhabi 129188, United Arab Emirates; rodrigo.torres@nyu.edu (R.T.); francesco.arneodo@nyu.edu (F.A.)
*   Correspondence: alessandro.re@unito.it

**Abstract:** At present, the use of non-destructive, non-invasive X-ray-based techniques is well established in heritage science for the analysis and conservation of works of art. X-ray fluorescence (XRF) plays a fundamental role since it provides information on the elemental composition, contributing to the identification of the materials present on the superficial layers of an artwork. Whenever XRF is combined with the capability of scanning an area to provide the elemental distribution on a surface, the technique is referred to as macro X-ray fluorescence (MA-XRF). The heritage science field, in which the technique is extensively applied, presents a large variety of case studies. Typical examples are paintings, ceramics, metallic objects and manuscripts. This work presents an uncommon application of MA-XRF analysis to furniture. Measurements have been carried out with the MA-XRF scanner of the INFN-CHNet collaboration at the Centro di Conservazione e Restauro "La Venaria Reale", a leading conservation centre in the field. In particular, a chinoiserie lacquered cabinet of the 18th century and a desk by Pietro Piffetti (1701–1777) have been analysed with a focus on the characterisation of decorative layers and different materials (e.g., gilding in the former and ivory in the latter). The measurements have been carried out using a telemeter for non-planar surfaces, and with collimators of 0.8 mm and 0.4 mm diameter, depending on the spatial resolution needed. The combination of the small measuring head with the use of the telemeter and of a small collimator has guaranteed the ability to scan difficult-to-reach areas with high spatial resolution in a reasonable time ($20 \times 10$ mm$^2$ with 0.2 mm step in less than 20 min).

**Keywords:** MA-XRF; conservation studies; furniture; Pietro Piffetti; chinoiserie lacquered cabinet

## 1. Introduction

X-ray fluorescence (XRF) is well established in the non-destructive, non-invasive analysis for the conservation, characterisation and prevention of works of art [1–5]. Whenever it is associated with the ability to scan an area, XRF provides the elemental composition related with the spatial distribution of the scanned area, and it is typically referred as macro X-ray fluorescence (MA-XRF) [1,4,6]. The MA-XRF technique is widely in use in the

heritage science field, and a number of research groups have been developing customized instruments to improve the quality of data obtainable with this technique [6–9].

The MA-XRF technique may be provided by portable instruments or by large-scale facilities like synchrotrons [10–12]. Despite their higher performances in terms of output beam parameters (higher intensity, smaller spot size), synchrotron facilities have the limitation of needing the artifacts to be transported to the laboratory, which is not always feasible.

Among others, the Cultural Heritage Network of the National Institute of Nuclear Physics (INFN-CHNet) [13–15] is engaged in the development of instruments, methods and techniques for applications in the heritage science field [16–19]. With this intent, the INFN-CHNet collaboration has developed its own MA-XRF scanner. The main characteristics of this device are its compactness, light weight, user-friendly interface, and versatility as described in the next section. It has already been employed for a number of different applications, including paintings [20–22], illuminated manuscripts [23], coins [24], mosaics [25] and ceramics [24].

The INFN-CHNet MA-XRF scanner has recently been employed on furniture. This further application is particularly interesting since, at the present time, the literature on XRF analysis on furniture is still poor [26–28], and no cases of MA-XRF analysis on furniture have been reported.

In this paper, the application of the INFN-CHNet MA-XRF scanner on two different pieces of furniture of the 18th century is presented. One is a chinoiserie wooden cabinet placed at the *Castello e Parco di Masino*, property of the *Fondo Ambiente Italiano* (FAI) in Piedmont. The other is a writing desk *Scrivania con scansia* by Pietro Piffetti, placed in the *Palazzo Chiablese* in Turin, Piedmont.

Both works of art have been studied in a specific framework at the Centro di Conservazione e Restauro (CCR) "La Venaria Reale". The lacquered cabinet was part of the research project "*Un ponte tra l'Oriente e il Piemonte*", a comparative study between lacquered oriental works of art of the 18th and 19th centuries, and their contemporary Western imitations. The main focus of the study was to determine the manufacturing techniques and materials employed in order to determine the elements required to date the manufactures and to distinguish between "original" oriental works of art and their Western replicas [29]. The study of the writing desk *Scrivania con scansia* by Pietro Piffetti from the *Palazzo Chiablese* has been realised within the research project on Pietro Piffetti carried out by the CCR [30,31]. This ongoing project is aimed at studying works of art by Pietro Piffetti and the Piedmontese cabinetmakers of the second half of the 18th century.

Thanks to the versatility of the INFN-CHNet MA-XRF scanner, it has been possible to study the decorative layers of the furniture, thus supporting their conservation processes and providing fundamental information to the research projects in which they are involved. The present work demonstrates the potentialities of the systematic use of a customized MA-XRF technique as a first approach to scientific studies on furniture.

## 2. Materials and Methods

The chinoiserie lacquered cabinet of the *Castello di Masino* (Figure 1b) is a typical example of furniture largely diffused throughout Europe in the second half of the 18th century. Historical research made it possible to date the furniture to 1780 and to attest to its production by a French workshop that was specialised in making furniture in imitation of oriental works assembling original Eastern lacquered panels. For this reason, it represents an important case study that permits the analysis of the aspects of both materials used in the Western imitations and those of the original oriental lacquers.

The writing desk *Scrivania con scansia* of *Palazzo Chiablese* (Figure 1a, $288 \times 155.5 \times 57.5$ cm$^3$) is a particularly significant work since it is the last documented masterpiece by Pietro Piffetti (Turin, 1701–1777), cabinetmaker of the Savoy court, dated to 1767. As part of the research on the Piedmontese cabinet-making, in particular on the furnishings by Pietro Piffetti, over the years, the CCR has been able to establish a database on the techniques

and materials used, which today makes it possible to accurately document the evolution of the history of furniture technology, the basis of these great multi-material masterpieces of international furniture.

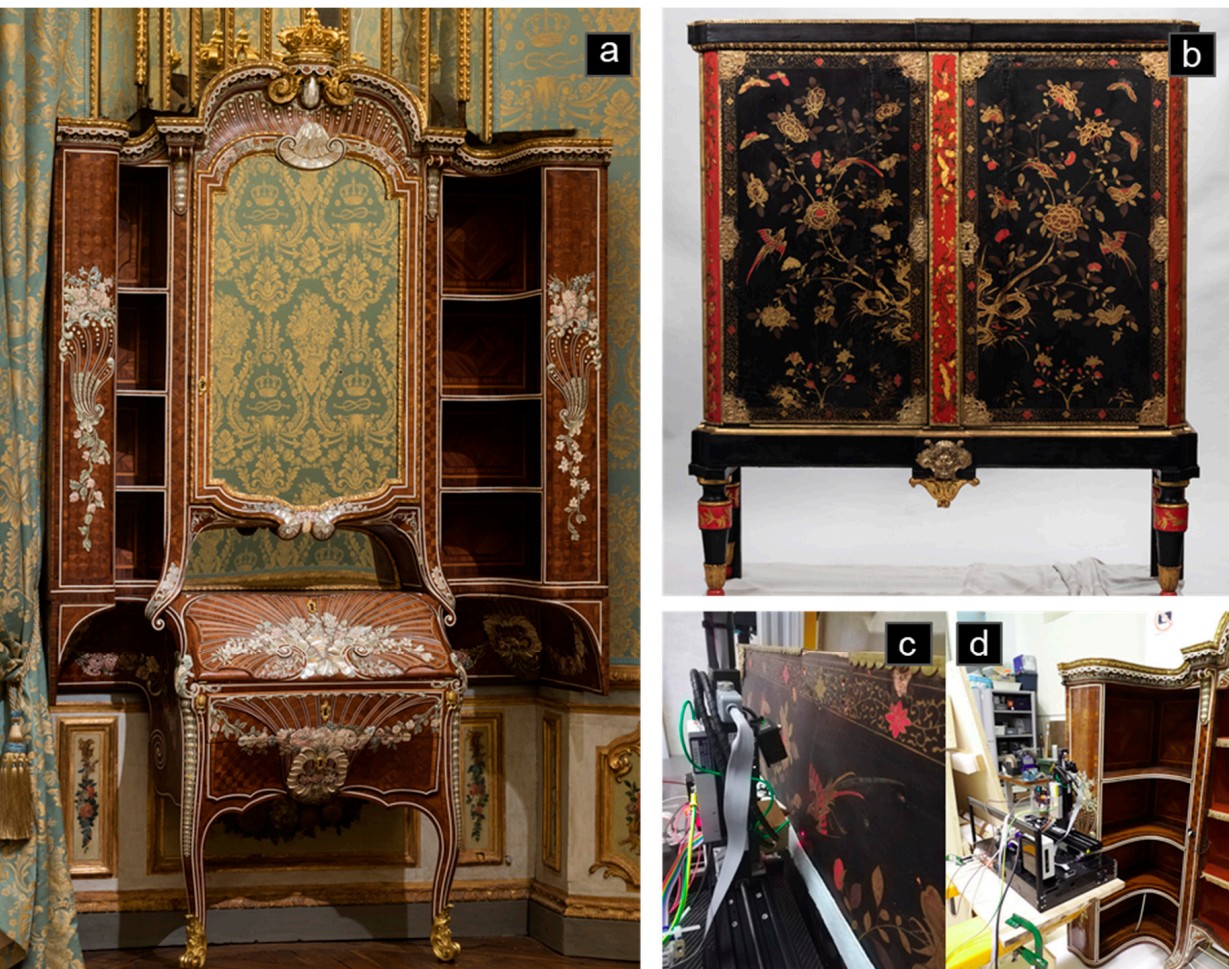

**Figure 1.** Pictures of the furniture and of the Cultural Heritage Network of the National Institute of Nuclear Physics (INFN-CHNet) macro X-ray fluorescence (MA-XRF) scanner during the measurements: (**a**) Writing desk *Scrivania con scansia* by Pietro Piffetti; (**b**) Chinoiserie wooden cabinet; (**c**) Scanner placed in front of the chinoiserie wooden cabinet; and (**d**) in front of the external side of the *scansia*.

The desk was created as a corner element of a boiserie inside the alcove hall of the *Palazzo Chiablese*. It is entirely inlaid and composed of veneers in violet rosewood and *bois de rose* and of polychrome engraved ivories and mother-of-pearl inlays. The cabinet-making furnishings of this period very rarely show inlays with polychrome engravings. Usually, they are made with black ink; hence, the possibility of mapping the nature of the inks and pigments present on this piece has provided an important element of comparison between the works that have been studied by the CCR in recent years [32,33].

The measurements have been carried out with the INFN-CHNet MA-XRF scanner at the CCR "La Venaria Reale". It is a compact ($60 \times 50 \times 50$ cm$^3$) and lightweight (around 10 kg) instrument completely assembled within the INFN-CHNet collaboration. Its main parts are the measuring head, composed by an X-ray tube (Moxtek© MAGNUM, 40 kV maximum voltage, 0.1 mA maximum anode current, Mo anode) with a collimator (changeable, typically between 400 μm and 2000 μm diameter), a silicon drift detector (Amptek© XR100 SDD, 50 mm$^2$ effective active surface, 140 eV FWHM at 5.9 keV), a telemeter (Keyence, model IA-100), three motor linear stages and a case containing all the electronics for acquisition and control. The motor stages (Physik Instrumente©, travel

ranges 30 cm horizontally, *x* axis; 15 cm vertically, *y* axis; and 5 cm towards the specimen, *z* axis) holding the measuring head are screwed onto the carbon-fibre case containing the electronic components and the power supplies. The maximum operating voltage is 40 kV. Signals are collected with a multichannel analyser (model CAEN DT5780, also inside the carbon-fibre case), and the whole system is controlled by a laptop. The control–acquisition–analysis software is developed within the collaboration and allows both on-line and off-line analysis.

The output of the MA-XRF analysis is a file containing the scanning coordinates and, for each position, the spectrum acquired. As a result, the counts are recorded for each position. For each scanned area, or a part of it, a single element can be selected by its energy transition value and shown as an elemental 2D map. For each peak, the energy range is manually selected around the centroid according to its FWHM. The relative intensity of each element in a map is shown in greyscale, in which the maximum intensity is in white and the minimum in black.

Furthermore, multi-elemental maps can be created. In those maps, different elements are displayed in different colours (red, green, blue). This option permits the association of one or more pigments, by means of their chemical elements, to the visible features, allowing an immediate spatial distribution of the pigments (see, for example, Section 3.1).

A scan is carried out on the *x* axis, and a step size of 1 mm is typically set on the *y* axis, resulting in a pixel size of 1 mm$^2$. However, both the measuring speed and the pixel size can be adjusted depending on the need, as reported in the case of the writing desk. This ability, combined with the changeable dimension of the collimator, allows optimisation of the measuring time, and thus the maintenance of an adequate spatial resolution.

During measurements, the distance between the specimen and the measuring head is kept constant by means of the telemeter. The distance is set to the value of 6 mm, allowing the detector to point at the area irradiated by the source and to maximise the covered solid angle. Therefore, the detection efficiency is also improved for non-flat surfaces, especially for low-energy X-rays subjected to higher absorption by the air.

Moreover, for the detection of low-energy X-rays, helium flow may be conveyed between the target and the detector system from a nozzle installed close by the detector.

With the set-up described above, maps of elements with atomic numbers higher than Sodium ($Z > 11$) are efficiently provided by the instrument. A full review on the instrument can be found in [34].

## 3. Results and Discussion

### 3.1. Chinoiserie Lacquered Cabinet

For the investigation of this furniture, two areas, one for each panel, have been studied: one area corresponding to a flower (Figure 2), the other area to a flying bird (Figure 3).

For the two measurements, helium flow has been used. The scanning parameters are reported in Table 1.

**Table 1.** Scanning parameters of the areas of the chinoiserie lacquered cabinet.

| Area | Dimension (mm$^2$) | Source Voltage (kV) | Anode Current (μA) | Scanning Speed (mm/s) | Collimator Diameter (μm) | Step Size (mm) |
|---|---|---|---|---|---|---|
| Flowers | 145 × 105 | 28 | 30 | 3 | 800 | 1 |
| Flying bird | 175 × 135 | 28 | 30 | 3 | 800 | 1 |

In the first area, the query was related to the possible presence of arsenic-based compounds in the transparent yellow buds and flowers. Due to the overlap of the As Kα transition (10,544 keV) with the Pb Lα (10,552 keV), and of the As Kβ (11,726 keV) with the Hg Lβ (11,823 keV), the presence of arsenic can not be ascertained by a single elemental map. Therefore, maps of the major overlapping transitions among arsenic, mercury, and lead have been made and are reported in Figure 2d–f. As can be seen, even though the Hg

Lβ/As Kβ transition shows a high intensity in the red flower and the transparent yellow buds (Figure 2d), the Lα of Hg clearly shows its presence only in the red flower (Figure 2e), indicating that the signals in the area of the transparent yellow buds come from arsenic. The presence of the Pb Lβ transition has not been detected in the spectrum, confirming this result.

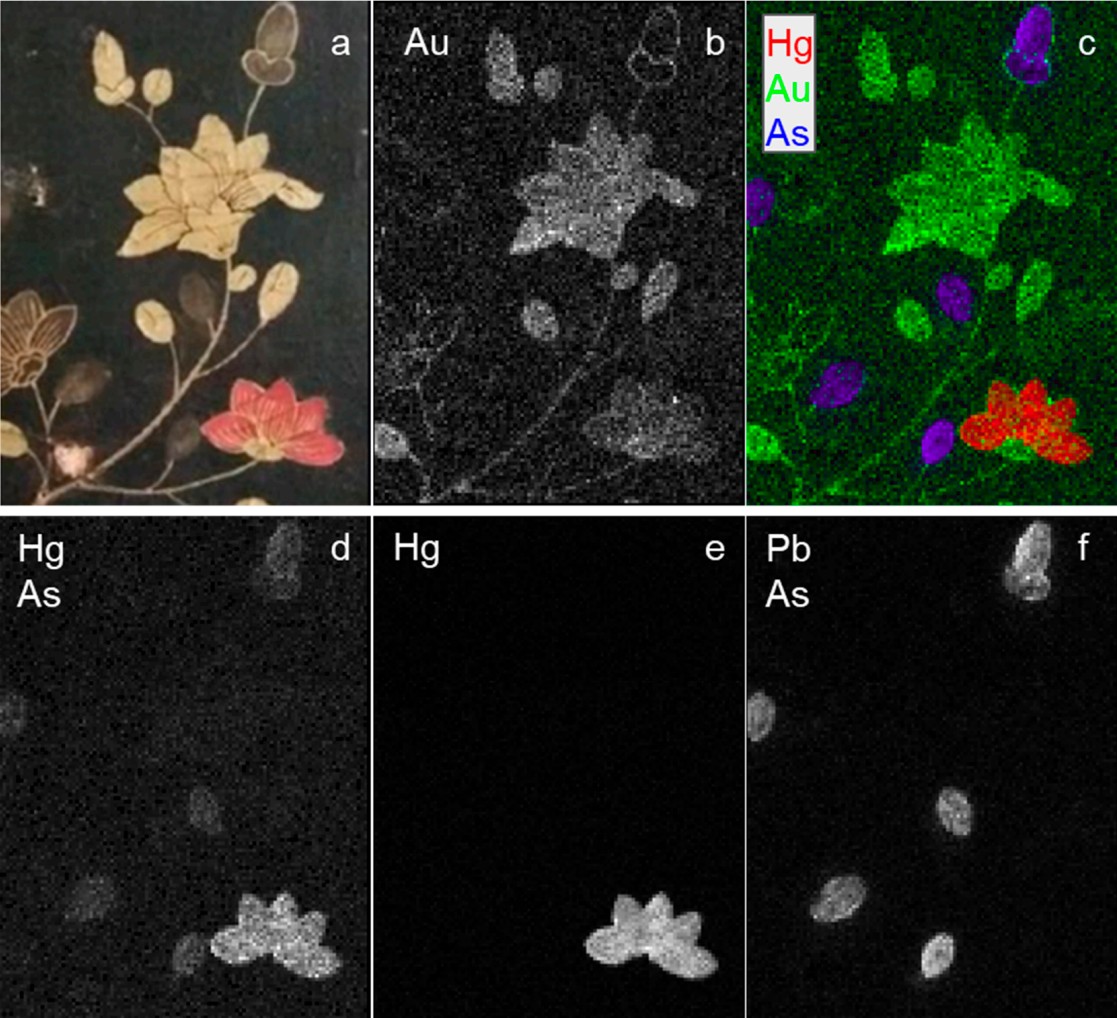

**Figure 2.** Elemental maps of the first area obtained by the INFN-CHNet MA-XRF scanner. (**a**) Visible. The maps presented are: (**b**) Au; (**c**) RGB map of Hg Lα (red), Au Lα (green) and As Kα (blue); (**d**) Hg Lβ/As Kβ transitions; (**e**) Hg Lα transition; (**f**) Pb Lα/As Kα transitions.

The presence of arsenic may be due to orpiment, but the use of realgar, pararealgar, or arsenic sulphide glass may not be excluded [35,36].

In order to clarify the composition of the different parts of the plant, an RGB map of Hg Lβ (red), Au Lα (green) and As Kα/Pb Lα (blue) transitions has been created and is reported in Figure 2c. As can be seen from Figure 2b, the bright yellow flowers, stems, outlines and highlights over red areas were made with gold. The red flower was likely realised with cinnabar-vermilion, of which the elemental composition is HgS [37].

The other area, even though it shows similarities with the previous, presents some essential differences in the decorative palette. The elemental maps of this area are shown in Figure 3. Compared with the area of the plant, the maps of mercury (Figure 3d) and gold (Figure 3g) attest to the probable use, respectively, of cinnabar-vermilion in the red tones, and of gold in the outline of the body and in one of the tails of the bird. The red tail is highlighted in gold as well.

As can be seen in the map of iron (Figure 3c), ochres-earths seem to have been used for the body of the bird. The eight black-blue feathers are related with a higher intensity of arsenic (Figure 3e), silicon (Figure 3h) and calcium (Figure 3b) compared with the rest of the area. In this case, with this technique it has not been possible to formulate a conclusive hypothesis on the decorative palette used by comparing the information with the present literature [38].

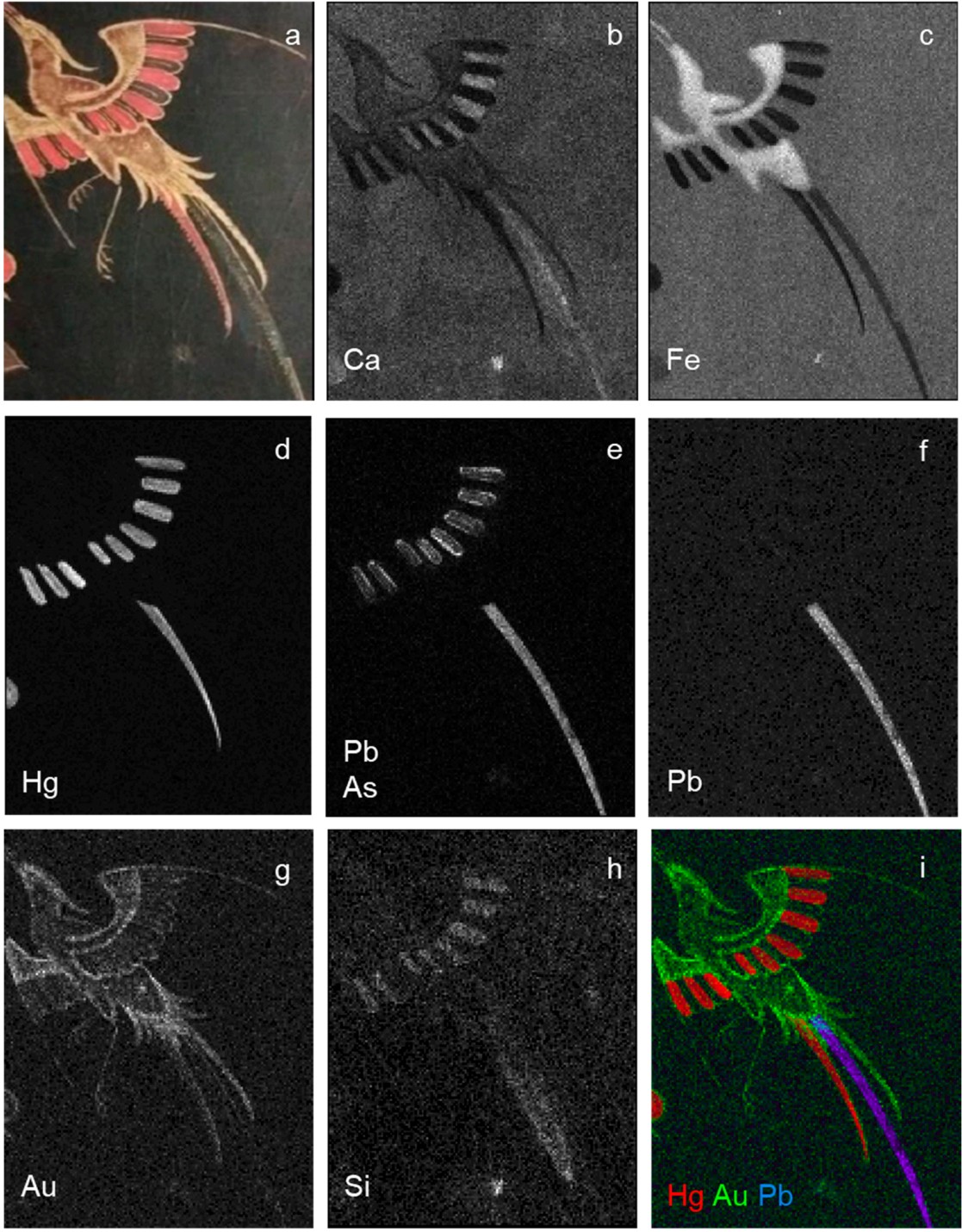

**Figure 3.** Elemental maps of the second area obtained by the INFN-CHNet MA-XRF scanner. (**a**) Visible. The maps presented are: (**b**) Ca Kα transition; (**c**) Fe Kα transition; (**d**) Hg Lα transition; (**e**) Pb Lα/As Kα transitions; (**f**) Pb Lβ transition; (**g**) Au Lα transition; (**h**) Si Kα transition; (**i**) RGB map of Hg (red), Au (green) and Pb (blue).

A different result has been achieved for the longest black tail, in which the presence of high Pb Lβ transition values reveals the use of Pb (Figure 3f). The literature [39,40] suggests that the presence of lead in black compounds may indicate the use of galena (PbS). The presence of sulphur can only be hypothesised since its transition (2.31 keV) overlaps with the transitions of mercury (M series) and lead (M series). However, with only XRF analysis, the use of another material like plattnerite ($PbO_2$) cannot be excluded. In addition, as can be seen from the map of silicon (Figure 3h), this element is present in the area of the tails, which may indicate a possible fourth tail with a similar composition to black-blue feathers. Finally, regarding the white small area below the red tail, the presence of a retouch is clearly visible in the maps of silicon (Figure 3h) and calcium (Figure 3b).

### 3.2. Writing Desk by Pietro Piffetti

The *Scrivania con scansia* by Pietro Piffetti has been analysed to characterise the decorating layers up on the marquetry. Three different areas were selected for this purpose: two areas on a drawer placed inside the writing desk, and one on a side, as shown in Figure 4. The choice was based on the hypotheses of different materials used and two possible conservation states inside and outside the writing desk.

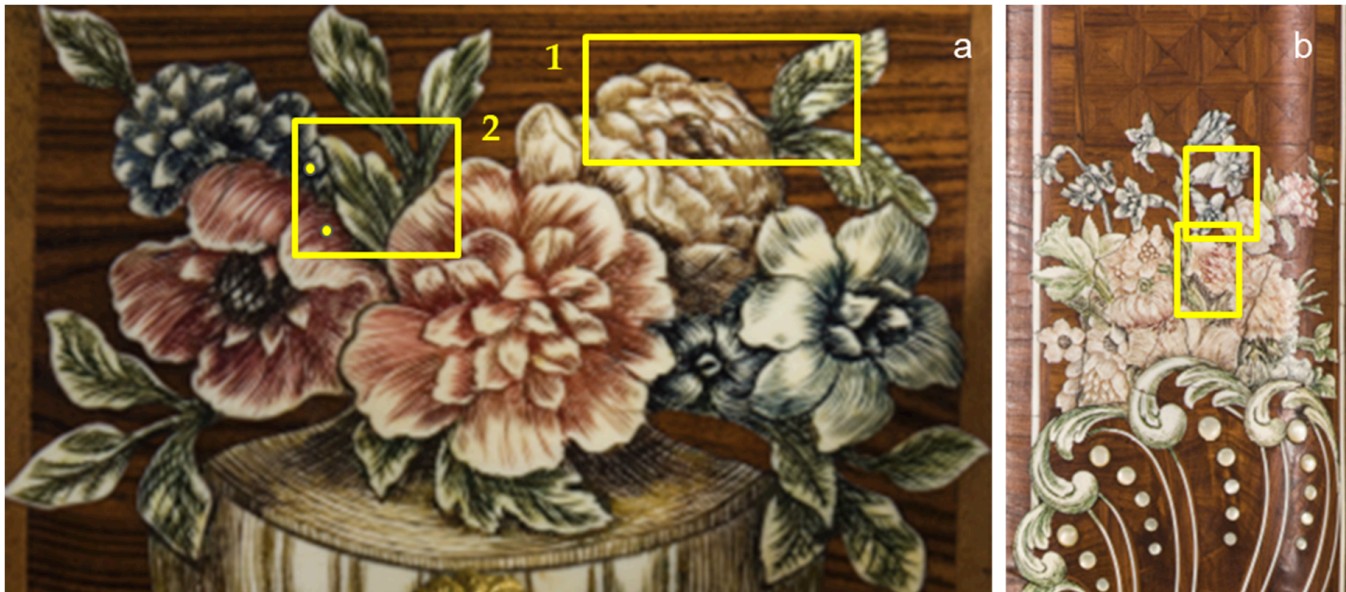

**Figure 4.** Selected areas for the measurements: (**a**) selected areas outlined in yellow and the two points in the black and red shades on the drawer; (**b**) particular of the external side. The scanned areas are outlined in yellow.

The scanning parameters are reported in Table 2. Because of the small size of the details of the decoration, a better spatial resolution compared to the lacquered cabinet was required. Since the scanner has an available set of different collimators, for this application the smallest one, with the diameter of 400 μm, has been used. Thanks to this feature, a higher spatial resolution has been achieved in the elemental maps.

**Table 2.** Scanning parameters of the areas analysed on the *Scrivania con scansia* by Pietro Piffetti.

| Area | Dimension (mm²) | Source Voltage (kV) | Anode Current (μA) | Scanning Speed (mm/s) | Collimator Diameter (μm) | Step Size (mm) |
|---|---|---|---|---|---|---|
| Drawer area 1 | 30 × 13 | 38 | 50 | 1 | 400 | 0.2 |
| Drawer area 2 | 20 × 10 | 28 | 50 | 1 | 400 | 0.2 |
| External side | 40 × 130 | 28 | 70 | 1 | 400 | 0.2 |

The elemental maps of the first area of the drawer are shown in Figure 5. From the maps of lead (Figure 5c,d) it can be seen that its presence is detected in correspondence of the white highlights of the flower, most likely due to the use of lead white [41].

The shading in the central part of the flower is related to the presence of iron (Figure 5g) and manganese (Figure 5h), which may lead to the hypothesis of the use of ochres-earths [39]. In the green leaves, high signals of copper (Figure 5f) and arsenic (Figure 5d,e) are clearly visible. Even though this can be explained by the presence of a copper-based pigment or dye mixed with an arsenic-based compound, the literature suggests that this combination is an unlikely possibility, whereas the green colour may be related to the use of emerald green or Scheele's green [39,42]. However, it is reported that the latter has only been used since 1814, excluding its use for this piece of furniture; therefore, it is plausible that emerald green was used for the green leaves.

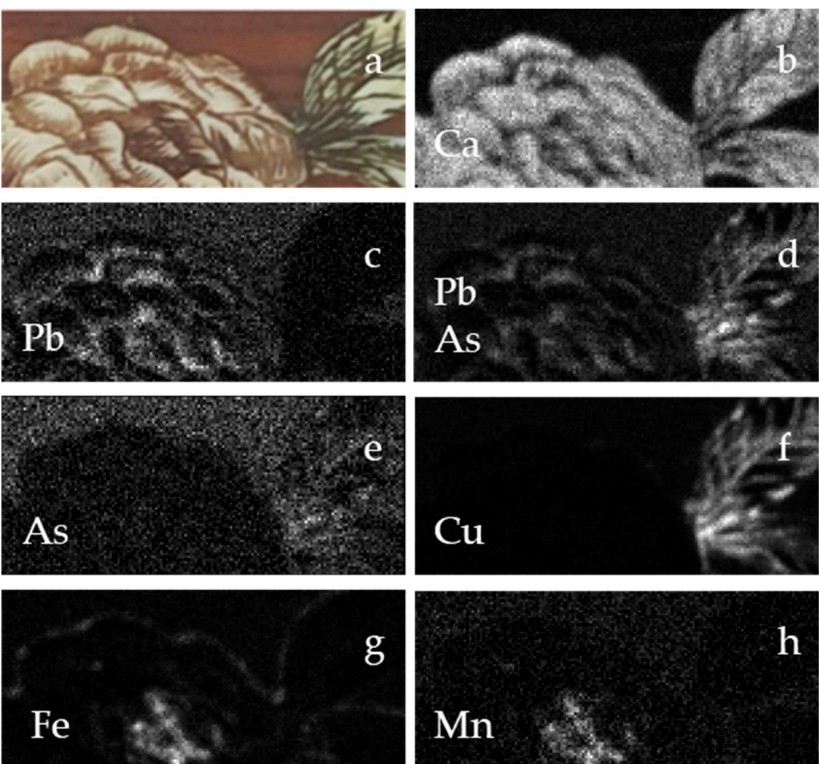

**Figure 5.** Elemental maps of the first area of the drawer obtained by the INFN-CHNet MA-XRF scanner. (**a**) Visible. The maps presented are: (**b**) Ca Kα transition; (**c**) Pb Lβ transition; (**d**) Pb Lα/As Kα transitions; (**e**) As Kβ transition; (**f**) Cu Kα transition; (**g**) Fe Kα transition; (**h**) Mn Kα transition.

The maps in the second area of the drawer, reported in Figure 6, confirm the same results of the first map, even though the black-blue flower on the top left shows a higher contribution of lead and iron. From the present literature, no single pigment is directly associated with the presence of those two elements [38,39]. Their presence can be explained by the use of Prussian blue combined with lead white, as reported in [43].

In the second area of the drawer shown in Figure 4, two points have been measured: one in the red tone and the other in the black tone. The measuring time was set to 120 s for each one to enhance the statistics of the spectra. As can be seen from Figure 7a, in the red tone the most intense peak is the calcium peak, probably related to the ivory beneath the decorative layer, whereas no other element with an atomic number above sodium was detected with relevant statistics. This result may indicate the use of organic dyes for this colour. The spectrum of the black-blue point (Figure 7b) shows a different elemental composition: the most prominent peaks are related to iron and lead (calcium is also present as a second peak in terms of intensity), confirming the result already discussed above.

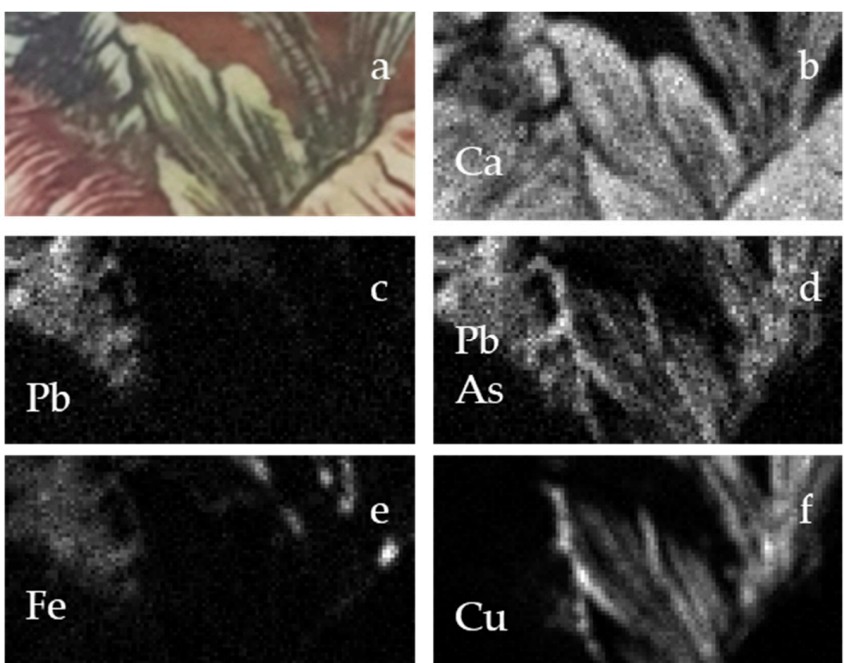

**Figure 6.** Elemental maps of the second area of the drawer. (**a**) Visible. The maps presented are: (**b**) Ca Kα transition; (**c**) Pb Lβ transition; (**d**) Pb Lα/As Kα transitions; (**e**) Fe Kα transition; (**f**) Cu Kα transition.

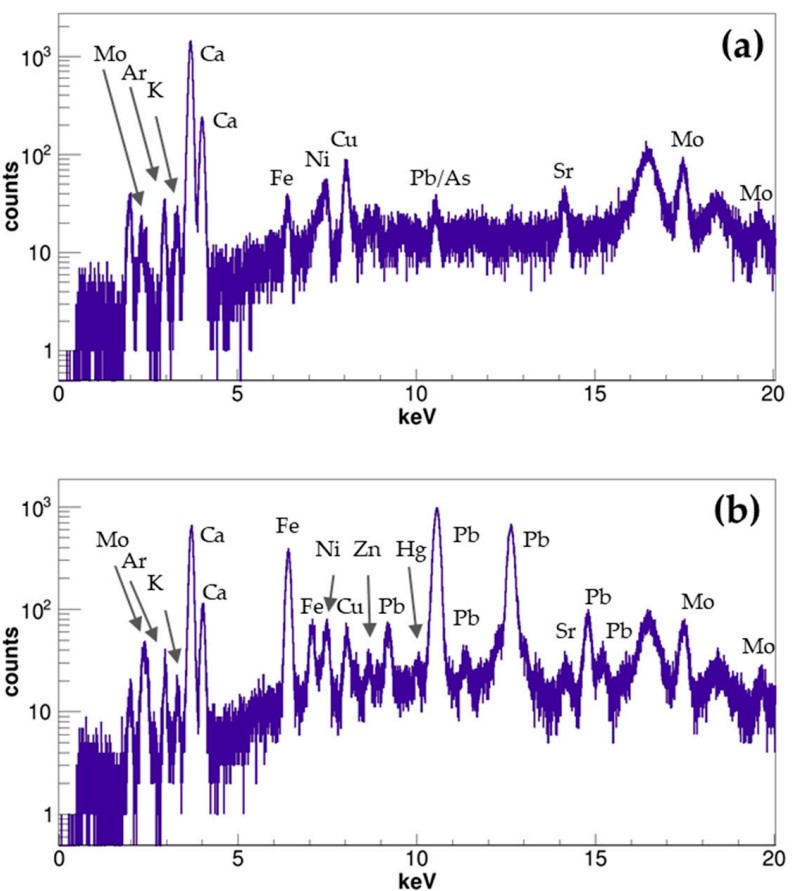

**Figure 7.** Spectra of the two points analysed with XRF: (**a**) red tone; (**b**) black tone. The presence of Ca probably due to ivory was detected in both.

On the external side, the two scans have been merged. As a result, elemental maps of the whole area indicated in Figure 4b were realised and are shown in Figure 8. In the area are green leaves, a dark blue-black stem and different flowers of red, black-blue and brown-orange tones. Some similarities with the drawer can be noted.

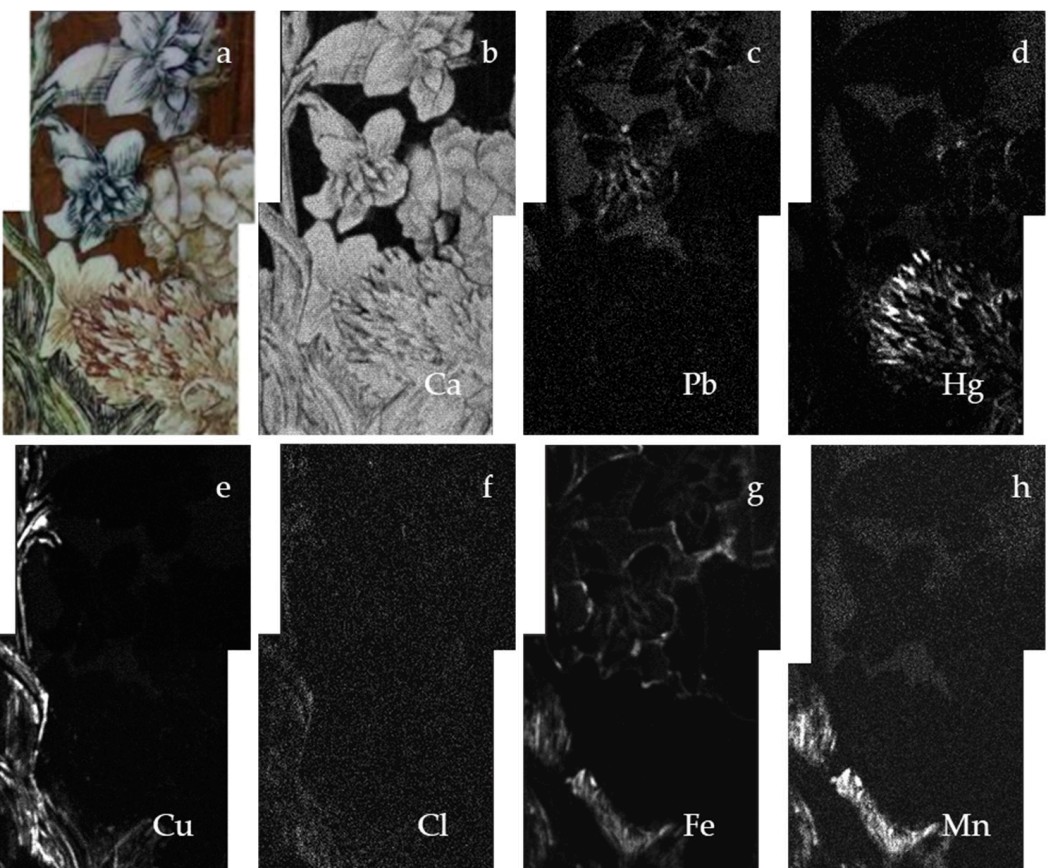

**Figure 8.** Elemental maps of the external area of the *Scrivania con scansia* obtained by the INFN-CHNet MA-XRF scanner. (**a**) Visible. The maps presented are: (**b**) Ca Kα transition; (**c**) Pb Lα transition; (**d**) Hg Lα transition; (**e**) Cu Kα transition; (**f**) Cl Kα transition; (**g**) Fe Kα transition; (**h**) Mn Kα transition.

In the shading of the red flower, mercury has been detected (Figure 8d), possibly due to the use of cinnabar-vermilion (HgS). Regarding the black-blue flowers, the same elements as in the black-blue flowers of the drawer—iron (Figure 8g) and lead (Figure 8c)—are present. Therefore, the decoration may have been realised with Prussian blue and lead white. In the blue-black stem area, the signals of iron and manganese are present in the elemental maps due to the probable use of iron-oxide-based compounds enriched with manganese oxides or other ochres-earths.

Regarding the dull-yellow flower, no map shows evidence of the presence of a characteristic element; therefore, the use of organic compounds can be hypothesised. Conversely, regarding the green parts in the drawer, a strong signal of copper (Figure 8e) associated with chlorine traces (Figure 8f) is present in the leaves on the side, whereas the presence of arsenic was not detected. This result led to the hypothesis of the use of different pigments in the green areas on the external side compared with the same colour in the drawer. Because the MA-XRF analysis does not allow the identification of compounds, it is not possible to determine which of the copper-based pigments has been used, or even if it was due to degradation effects as reported in [44].

A summary of the pigments hypothesised by the use of MA-XRF analysis is reported in Table 3.

**Table 3.** Summary of the hypothesised materials employed in the different areas.

| Area Scanned | Colour | Elements Detected | Materials Hypothesised |
|---|---|---|---|
| Chinoiserie cabinet—flowers | red flower<br>bright yellow flowers<br>transparent yellow buds | Hg<br>Au<br>As | cinnabar-vermilion<br>gold<br>arsenic-based compound |
| Chinoiserie cabinet—flying bird | red tail and feathers<br>bright yellow outlines<br>buff body<br>black-blue feathers<br>black tail | Hg<br>Au<br>Fe<br>As, Si, Ca<br>Pb | cinnabar-vermilion<br>gold<br>ochres-earths<br>?<br>galena |
| Writing desk—drawer area 1 | green leaves<br>dark rust tone of the flower<br>rust tone of the flower<br>white highlights | As, Cu<br>Fe, Mn<br>-<br>Pb | emerald green<br>ochres-earths<br>organic compounds<br>lead white |
| Writing desk—drawer area 2 | black-blue flower<br>green leaves | Pb, Fe<br>As, Cu | Prussian blue and lead white<br>emerald green |
| Writing desk—external side | green leaves<br>black-blue flower<br>red flower<br>blue-black dark stem<br>dull yellow flower | Cu, Cl<br>Pb, Fe<br>Hg<br>Fe, Mn<br>- | copper-based compound<br>Prussian blue and lead white<br>cinnabar-vermilion<br>ochres-earths<br>organic compounds |

## 4. Conclusions

The use of the MA-XRF technique on furniture has provided information on the elemental spatial distribution of the decorative layers. In particular, the INFN-CHNet MA-XRF scanner, due to its versatility in terms of hardware adaptability (different size of collimators, use of the telemeter) and in terms of software settings (scanning speed and step size during the measurements, data saving for an off-line analysis) has proved to be suitable for studies on this genre of works of art. A number of results have been achieved in identifying materials in the decorative layer. In both cases, the data on the polychromatic surfaces have provided information on the materials used: for instance, the use of arsenic-based compounds in the chinoiserie cabinet and the use of different pigments in the engraved ivories of the writing desk by Pietro Piffetti. However, the detection of only the elemental composition is a strong limitation for this kind of study. Therefore, the use of the technique is suggested as a first approach, while a multi-technical study is advisable for the identification of the compounds, as reported in [28]. For this reason, the INFN-CHNet group is involved in the development of instruments and techniques to support conservation processes, characterisation, and dating techniques in the heritage science field.

**Author Contributions:** Conceptualization, F.T., A.R. and A.L.G.; Data curation, L.S., L.G. (Laura Guidorzi), A.M. and C.R.; Formal analysis, L.S., L.G. (Laura Guidorzi), A.M. and C.R.; Funding acquisition, F.T. and A.L.G.; Investigation, L.G. (Laura Guidorzi), C.R., L.C., M.N., R.T. and A.R.; Methodology, L.G. (Laura Guidorzi), C.R., L.C., C.C., L.G. (Lorenzo Giuntini), M.M., F.T., M.N., R.T., F.A. and A.R.; Resources, M.N. and S.D.B.; Software, L.C., C.C. and F.T.; Supervision, A.R.; Visualization, L.S., L.G. (Laura Guidorzi), A.M. and C.R.; Writing—original draft, L.S.; Writing—review and editing, L.S., L.G. (Laura Guidorzi), A.M., A.R. and A.L.G. All authors have read and agreed to the published version of the manuscript.

**Funding:** The research was funded within the INFN-CHNet project. This project has received funding from the European Union's Horizon 2020 research and innovation programme under the Marie Skłodowska-Curie grant agreement No 754511 (PhD Technologies Driven Sciences: Technologies for Cultural Heritage—T4C).

**Institutional Review Board Statement:** Not applicable.

**Informed Consent Statement:** Not applicable.

**Data Availability Statement:** Not applicable.

**Acknowledgments:** The authors wish to warmly thank Marco Manetti for his suggestions, advice and invaluable technical support.

**Conflicts of Interest:** The authors declare no conflict of interest.

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
