# Peer review of "The Importance of Being Versatile: INFN-CHNet MA-XRF Scanner on Furniture at the CCR “La Venaria Reale”"

_applsci, doi:10.3390/app11031197_

Round 1

Reviewer 1 Report

Overall, very strong work.  The authors noted the spectral energy overlaps of various elements and discussed in a clear manner how the process they used to discern the pigments.  I would suggest adding a sentence about the sensitivity of open-architecture XRF units in regards to light elements in ambient conditions (I am assuming there is no helium flow in the set-up) and also how the air gap between the scanner head to the analyzed surface impacts this detection sensitivity.  This is another reason why sulfur (and other light Z elements) is difficult to observe unless there is a significant amount of it.

It is difficult to see in the images provided, but did you analyze any furniture surface that was curved/had significant topography?  Based on the images in the manuscript, possibly Fig. 1d (on the right)?  If so, what was the protocol that you implemented to deal with a non-flat surface?

Reviewer 2 Report

This is a compact paper that is well structured with clear hypothesis.

The primary novelty of the paper is the use of the an MA-XRF instrument on antique furniture, and in that regard I will take the authors word that this is a novel use of this instrument and XRF imaging.

Overall I recommend this paper for publication after hopefully some minor edits, clarifications and additions.

Introduction

Of course, a full literature review is beyond the scope of this paper, and multiple citations are provided. I would however like to see a sentence or two with accompanying citations about other forms of MA-XRF, such as the use of synchrotron radiation (which I admit I am biased as I work at a synchrotron and have performed analyses on multiple cultural heritage objects), and potentially mentioning the pros and cons. For example, the nature of synchrotron radiation (such as orders of magnitude higher flux than an x-ray tube) provides greater sensitivity and smaller x-ray spot sizes for higher resolution. But the ability to analyze pieces such as furniture is almost not possible due to the experimental constraints. See for example:

U. Bergmann, L. Bertrand, N.P. Edwards, P.L. Manning, R.A. Wogelius (2019) Chemical Mapping of Ancient Artifacts and Fossils with X-Ray Spectroscopy in Synchrotron Light Sources and Free-Electron Lasers: Accelerator Physics, Instrumentation and Science Applications. Eds: E.J. Jaeschke, et al. Springer Nature.  DOI: 10.1007/978-3-319-04507-8_77-1.

Experimental

The instrument used is clearly explained and scanning parameters for each data set are shown. There are some clarifications I would like to see.

Line 122: The output of the MA-XRF analysis is a file containing the scanning coordinates and, for each position, the spectrum acquired.

Please clarify if each pixel of the map files contains an XRF spectrum.

Line 123: For each scanned area, or a part of it, a single element can be selected by its energy transition value,

How are these elements selected? I assume that a binned range of emission energy is selected, say plus or minus 20eV or so from the known emission energy?

Line 136: During the measurements, the distance between the specimen and the measuring head is kept constant by means of the telemeter. 

This is a nice touch as it removes changes in signal from variances on sample to instrument distance. However, these distances are not listed in the experimental parameter tables in the results section. Please add.

Line 137: Maps of elements with atomic numbers higher than Sodium (Z >11) are efficiently provided by the instrument. 

Ok…. But later you mention elements such as sulfur. These are hard to get a good signal in air due to absorption of these x-rays by air. How was this issue mitigated? This also relates to the sample distance question above. Only a few mm of air would significantly reduce the signal for lighter elements.

Line 145: For the two measurements helium flow has been used. 

What does this mean? Does this mean that a helium flow was added between the object and the detector in order to reduce absorption of lower energy x-rays from lighter elements and thus improve the signal from those elements?

Other notes

Why were such high incident x-rays used? Were you expecting to detect/excite K or L edges of heavier elements? Or is this just the normal efficient operating energy of the instrument? Would this instrument have more flux if a lower incident energy was used?

For the overlapping emission lines, if indeed a spectrum was collected for every pixel, there are at least a couple of freely available software packages that can fit the spectra and deconvolve these overlapping lines. If there is only raw count data per pixel, the authors are correct and it is indeed difficult to deconvolve these elements. 

Results

There are some inconsistencies with the reporting and the assignment of pigments to the presence of certain elements. For example,

Line 190: The literature [35,36] suggests that the presence of lead in black compounds may indicate the use of galena (PbS). 

This is a good statement with citations. But also present in the manuscript are a few occasions where pigment assignments are not accompanied by citations, for example,

Line 168: The red flower has been likely realised with cinnabar-vermilion.

For the unitiated who might be reading this paper, what element is expected in cinnabar? Please correct these inconsistancies and add citations to each of these statements. 

Line 178 - regarding sulphur and silicon. The authors themselves allude to the ambiguity of the data for these elements, and this is made all the more difficult to swallow by not presenting the data. If the authors themselves cannot use these data with confidence then they should not be mentioned. Again, some clarification of the the absorption issue with the instrument of the sample distance and atmosphere (air or helium) would be useful.

Figure 5: There are clearly other elements than the ones mentioned or labelled. I believe that there is significant copper and zinc peaks. These spectra might be clearer if presented as a logarithmic scale to emphasize the smaller peaks.

Other notes

There are a few of instances where something is “not reported”. e.g. line 176 and 178. I am not comfortable with this. Any result that is reported must have the appropriate data presented, otherwise we are just taking the authors word for it and we cannot interrogate it ourselves. Please either add the appropriate data or remove the statements.

Conclusions

This manuscript taken as a techniques study is quite nice and could be if interest to an audience outside the field, and the conclusions are appropriate if taken in that sense. The novelty of analyzing furniture (if true) is worthy and interesting enough for publication.

However, the set up in the introduction, particular in the paragraph starting at line 63, is not mentioned at all in the conclusion.

“The main focus of the study was to determine the manufacturing techniques and the materials employed in order to find out elements to date the manufactures and to distinguish between the “original” oriental works of art and their western replicas.”

How these results and the use of instruments such as MA-XRF in the future might fit into and supplement the larger studies of the objects is not mentioned. After the introduction, I was expecting some insight into these aspects and that these data had contributed in some way or corroborated existing hypotheses. Instead the conclusion is quite vague and generic. This should be corrected before publication.
